# A Molecular Electron Density Theory Study of the [3+2] Cycloaddition Reaction of an Azomethine Ylide with an Electrophilic Ethylene Linked to Triazole and Ferrocene Units

**DOI:** 10.3390/molecules27196532

**Published:** 2022-10-03

**Authors:** Luis R. Domingo, Mar Ríos-Gutiérrez, Assem Barakat

**Affiliations:** 1Department of Organic Chemistry, University of Valencia, Dr. Moliner 50, 46100 Burjassot, Valencia, Spain; 2Department of Chemistry, College of Science, King Saud University, P.O. Box 2455, Riyadh 11451, Saudi Arabia

**Keywords:** Molecular Electron Density Theory (MEDT), [3+2] cycloaddition (32CA), azomethine ylide (AY), spirooxindoles, ferrocene, triazole

## Abstract

The [3+2] cycloaddition (32CA) reaction of an azomethine ylide (AY) with an electrophilic ethylene linked to triazole and ferrocene units has been studied within the Molecular Electron Density Theory (MEDT) at the *ω*B97X-D/6-311G(d,p) level. The topology of the electron localization function (ELF) of this AY allows classifying it as a *pseudo(mono)radical* species characterized by the presence of two monosynaptic basins, integrating a total of 0.76 e, at the C1 carbon. While the ferrocene ethylene has a strong electrophilic character, the AY is a supernucleophile, suggesting that the corresponding 32CA reaction has a high polar character and a low activation energy. The most favorable *ortho*/*endo* reaction path presents an activation enthalpy of 8.7 kcal·mol^−1^, with the 32CA reaction being exergonic by −42.1 kcal·mol^−1^. This reaction presents a total *endo* stereoselectivity and a total *ortho* regioselectivity. Analysis of the global electron density transfer (GEDT) at the most favorable **TS-on** (0.23 e) accounts for the high polar character of this 32CA reaction, classified as forward electron density flux (FEDF). The formation of two intermolecular hydrogen bonds between the two interacting frameworks at the most favorable **TS-on** accounts for the unexpected *ortho* regioselectivity experimentally observed.

## 1. Introduction

The construction of enantiomerically pure compounds has experienced a significant progress during recent decades by asymmetric reactions using organocatalysis, chiral susbtrates/auxiliaries, and reagents able to control the stereochemistry of desired enantioselective molecules. To synthesize pharmacologically active chiral molecules for specific receptors or proteins, the establishment of structure−function relationships and mechanistic studies are indispensable. The [3+2] cycloaddition (32CA) reactions, which have proved to be highly selective, efficient, environment-friendly and atom-economical, are among the most efficient asymmetric reactions to construct pharmacologically active chiral molecules in a regio- and/or stereoselective fashion [1,2,3,4].

Organic chemists have been extensively made great efforts to understand the electronic structure of three-atom-components (TACs) generated in situ as intermediates in 32CA reactions. Based on the recently proposed Molecular Electron Density Theory (MEDT) [5], four different types of TACs, namely zwitterionic (zw), carbenoid (cb), *pseudo(mono)radical* (pmr), and *pseudodiradical* (pdr) TACs, and their reactivity towards ethylene have been characterized [6]. Notably, *pseudo(mono)radical* and *pseudodiradical* TACs are very reactive due to their instability. These electronic structures and the corresponding reactivities can be modified by substitution.

Pyrrolidines are five-membered heterocyclic compounds used as pharmacologically relevant scaffolds in drug design [7,8,9], which can be easily constructed by 32CA reaction of AYs with olefins (see Figure 1). On the other hand, the introduction of a spiro-ring in heterocyclic compounds is a widely used strategy in drug design to provide additional conformational restriction. Thus, the 32CA reaction of exocyclic AY **4** generated from isatin, i.e., a dicarbonyl compound, yields spirooxindoles **6**, which possesses significant pharmacological properties [10,11,12,13,14].

MEDT studies of the 32CA reactions of the simplest AY **7,** CH_2_-NH-CH_2_ [15], and carbonyl ylide **8**, CH_2_-O-CH_2_ [16], have shown that the presence of a *pseudoradical* center at each one of the two methylene carbons of these TACs causes the corresponding *pdr-type* 32CA reactions with non-activated ethylenes to have an unappreciable activation energy, lower than 1.0 kcal·mol^−1^. However, the presence of an electron-releasing phenyl group and electron-withdrawing (EW) carboxyl –CO_2_R or nitrile –CN groups at the two methylenes of the simplest AY **7**, just as AY **1**, stabilizes its *pseudodiradical* electronic structure, modifying the experimental reactivity of these substituted AYs to that of *pseudo(mono)diradical* [17], carbenoid [18], or even zwitterionic TACs [19].

Recently, Barakat and co-workers experimentally studied the synthesis of spirooxindoles **11** and **12** via 32CA reactions of AYs **9**, generated from substituted isatins and secondary amines, with disubstituted olefins, in excellent regio- and stereoselectivity (see Figure 2) [20,21,22]. The plausible reaction mechanism was suggested to take place via an *endo* stereoisomeric path in which the substitution of the olefin derivatives **10** plays a crucial role in the regioselective formation of the products **11** or **12**.

Thus, the 32CA reaction of AY **13** with phenyl vinyl sulphone **14** yields the spirooxindole **15** with total *meta* regio- and *endo* stereoselectivity (see Figure 3) [23]. A MEDT study of the 32CA reaction of model AY **16** with phenyl vinyl sulphone **14** [24] showed that this 32CA reaction takes place via a *two-stage one-step* [25] mechanism involving a highly asynchronous transition-state structure (TS), with a high *endo* stereoselectivity and high *meta* regioselectivity. The reaction presents a very low activation enthalpy, 5.7 kcal·mol^−1^, as a consequence of the strong polar character of the reaction; the global electron density transfer (GEDT) [26] at the most favorable *meta*/*endo* TS was 0.31 e. This behavior is a consequence of the supernucleophilic character of AY **16** and the strong electrophilic character of vinyl sulphone **14**. The *meta* regioselectivity was explained by the more favorable two-center interaction between the most nucleophilic center of AY **16**, the C1 carbon, and the more electrophilic center of phenyl vinyl sulphone **14**, the C4 carbon, anticipated by the analysis of the Parr functions [27].

The ferrocene scaffold is an interesting organometallic architecture with a diversity of applications in medicine, photochemistry, as well as a building block for many organic synthetic transformations [28,29,30,31]. Many synthesized or naturally occurring organic compounds incorporating the ferrocene unit possess pharmacological activity which is sold in the market or in more advanced preclinical stages. Introducing the ferrocene synthon into the spirooxindoles scaffold is challenging for experimental chemists.

Triazoles, in particular the 1,2,3-triazole motif, have attracted great deal of attention due to their biological activities as anti-malarial agents, carbonic anhydrase inhibitors, agents for the tuberculosis treatment, etc., [32,33,34,35,36,37,38,39,40,41,42,43,44]. Introducing another interesting pharmacophore such as the 1,2,3-triazole framework into the spirooxindoles in combination with the ferrocene organometallic unit may lead to particular properties that could be useful for different applications in varied fields such as supramolecular chemistry, biochemistry, biosensing probes, or conducting polymer chemistry.

Very recently, Barakat et al. have experimentally reported the 32CA reaction of AY **18** with ethylene derivative **19**, in the synthesis of spirooxindoles **20**, with high *ortho* regio- and *endo* stereoselectivity (see Figure 4) [45]. Interestingly, this 32CA shows an opposite regioselectivity to that shown in the reaction with vinyl sulphone **14** (see Figure 3).

Herein, the 32CA reaction of AY **21** with ferrocene ethylene **22** yielding spirooxindole **23**, as a computational model of the 32CA reaction studied by Barakat et al., is theoretically studied within the MEDT in order to understand the behavior of the 32CA reactions involving ferrocene ethylene derivatives (see Figure 4) and the origin of the unexpected *ortho* regioselectivity.

## 2. Results and Discussion

The present MEDT study has been divided in four sections: (i) first, an ELF topological analysis at the ground state of AY **21** and ferrocene ethylene derivative **22** is performed; (ii) in the second part, the conceptual DFT (CDFT) reactivity indices at the ground state of the reagents are analyzed; (iii) in the third part, the competitive reaction paths associated with the 32CA reaction of AY **21** with ferrocene ethylene **22** are studied; and (iv) the origin of the *ortho* regioselectivity is finally analyzed.

### 2.1. ELF Topological Analysis at the Ground State of AY **21** and Ferrocene Ethylene Derivative **2****2**

The topological analysis of the electron localization function (ELF) [46] at the ground state allows a quantitative and qualitative description of the electronic structure of organic molecules [47]. Given the structure−reactivity relationship found in TACs [6], an ELF topological analysis of AY **21** was first performed in order to characterize its electronic structure and gain some insight about its reactivity. The most significant ELF basin attractor positions and valence basin populations of AY **21** and ferrocene ethylene **22** are given in Figure 1.

The ELF of AY **21** shows the presence of two monosynaptic basins, V(C1) and V’(C1), integrating a total of 0.77 e, one V(C1,N2) disynaptic basin integrating 2.29 e and one V(N2,C3) disynaptic basin integrating 3.43 e, characterizing the C1–N2–C3 AY core. While the V(C1,N2) disynaptic basin is associated with a C1–N2 single bond, the V(N2,C3) disynaptic basin is associated with a somewhat underpopulated N2–C3 double bond. The presence of the two monosynaptic basins at the C1 carbon integrating less than 1.0 e, which are associated with a *pseudoradical* C1 center [15], allows the classification of AY **21** as a *pseudo(mono)radical* TAC participating in *pmr-type* 32CA reactions [6].

On the other hand, the ELF of ferrocene ethylene derivative **22** shows the presence of two disynaptic basins, V(C4,C5) and V’(C4,C5) integrating a total of 3.32 e, a V(C5,C6) disynaptic basin integrating 2.29 e, one V(C6,O7) disynaptic basin integrating 2.25 e, and two monosynaptic basins, V(O7) and V’(O7) integrating a total of 5.35 e. While the two V(C4,C5) and V’(C4,C5) disynaptic basins are associated with an underpopulated C4–C5 double bond, the V(C6,O7) disynaptic basin is associated with a carbonyl C6–O7 single bond, resulting from a strong polarization of the C6–O7 bonding region towards the electronegative O7 oxygen, which shows a non-bonding region with a high electron density.

Natural Population Analysis (NPA) [48,49] of the charge distribution shows that the two reactive carbons of AY **21** are negligibly charged by less than ± 0.08 e while the N2 nitrogen is negatively charged by −0.26 e. Interestingly, the natural charges of ethylene derivative **22** indicate that the C4 and C5 carbons are negatively charged by −0.32 and −0.09 e, respectively while the carbonyl C6 carbon is strongly positively charged by 0.53 e as a consequence of the strong polarization of the carbonyl C6–O7 bonding region towards the electronegative O7 oxygen, which has a negative charge of −0.60 e.

### 2.2. CDFT Analysis at the Ground State of the Reagents

The reactivity indices defined within CDFT [50,51] have shown to be powerful tools to understand the reactivity in polar reactions [52]. The global reactivity indices, namely, the electronic chemical potential *μ*, chemical hardness *η*, electrophilicity ω, and nucleophilicity *N* indices, of AY **21** and ferrocene ethylene **22** are gathered in Table 1.

The electronic chemical potential [53] of AY **21** with *μ* = −2.79 eV is higher than that of ferrocene ethylene **22** with *μ* = −3.53 eV, indicating that along a polar 32CA reaction the GEDT [26] will take place from AY **21** to ferrocene ethylene **22**, the corresponding polar 32CA reaction being classified as of forward electron density flux (FEDF) [54].

AY **21** presents an electrophilicity ω index [55] of 1.17 eV, being classified as a moderate electrophile within the electrophilicity scale [51], and a nucleophilicity *N* index [56] of 4.67 eV, being classified as a strong nucleophile within the nucleophilicity scale [51]. The strong nucleophilic character of AY **21**, higher than 4.0 eV, allows its classification as a supernucleophile [57]. On the other hand, ferrocene ethylene **22** presents an electrophilicity ω index of 1.72 eV, being classified as a strong electrophile, and a nucleophilicity *N* index of 3.78 eV, being also classified as a strong nucleophile. The supernucleophilic character of AY **21**, together with the strong electrophilic character of ferrocene ethylene **22**, indicates that the corresponding 32CA reaction will have a high polar character, being classified as FEDF [54].

In a polar 32CA reaction involving non-symmetric species, the most favorable reaction path is that involving the two-center interaction between the most electrophilic and the most nucleophilic centers of the two reagents [58]. Many studies have shown that the electrophilic P_k_^+^ and nucleophilic P_k_^−^ Parr functions [27], resulting from the excess of spin electron density gathered via the GEDT, are among the most accurate tools for the analysis of the local reactivity in polar and ionic processes [52]. Hence, according to the characteristics of the reagents, the nucleophilic P_k_^−^ Parr functions of AY **21** and the electrophilic P_k_^+^ Parr functions of ferrocene ethylene **22** were analyzed (see Figure 2).

The two C1 and C3 carbons of AY **21** are nucleophilically activated by P_k_^−^ = 0.38 and 0.32, respectively, with the C1 carbon being the most nucleophilic center of the TAC. Note that the N2 nitrogen is nucleophilically deactivated (P_k_^−^ = −0.12). On the other hand, the *β*-conjugated C4 position of ferrocene ethylene **22** is the most electrophilically activated of this species (P_k_^+^ = 0.25).

Based on the analysis of the Parr functions, the *meta* regioisomer is expected to be the preferred one as a consequence of the slightly higher nucleophilic activation of the C1 carbon of AY **21** than that of the C3 one.

### 2.3. Analysis of the Competitive Reaction Paths Associated with the 32CA Reaction of AY **21** with Ferrocene Ethylene **22**

In order to determine the mechanism of this 32CA reaction, the competitive reaction paths associated with the 32CA reaction of AY **21** with ferrocene ethylene **22** were analyzed. Due to the non-symmetry of both reagents, two pairs of *endo* and *exo* stereoisomeric reaction paths and two pairs of *ortho* and *meta* regioisomeric ones are possible (see Figure 5). The analysis of the stationary points found along with these reaction paths indicates that this 32CA reaction takes place through a one-step mechanism. The *ω*B97X-D/6-311G(d,p) relative electronic energies, in gas phase and in methanol, are given in Table 2. Total electronic energies are given in Appendix A in Appendix A.

An exhaustive analysis of the potential energy surface associated with this 32CA reaction allowed finding a series of molecular complexes (MCs), which are in equilibrium, in an early stage of the reaction. The most favorable **MC-on**, which opens the *ortho*/*endo* reaction path, is found to be 25.6 kcal·mol^−1^ more stable than the separated reagents (see Figure 5).

Some appealing conclusions can be obtained from the analysis of the relative energies of the stationary points involved in this 32CA reaction: (i) the most favorable **TS-on** is found to be 17.1 kcal·mol^−1^ below the separate reagents; (ii) however, when the formation of **MC-on** is considered, the activation energy of this 32CA reaction becomes positive at 8.5 kcal·mol^−1^; (iii) this 32CA reaction is strongly exothermic by −49.9 kcal·mol^−1^. Consequently, the formation of the experimental spirooxindole **23** takes place by kinetic control. Note that the reaction is strongly exergonic by −22.4 kcal·mol^−1^ (see later); (iv) this 32CA reaction is completely *ortho* regioselective, as **TS-mn** is found to be 5.3 kcal·mol^−1^ above **TS-on**; and (v) this 32CA reaction is completely *endo* stereoselective, as **TS-ox** is found to be 8.3 kcal·mol^−1^ above **TS-on**.

Inclusion of solvent effects of methanol stabilizes all species by between 15.3 and 19.7 kcal·mol^−1^; the reagents are the most stabilized [59]. As a consequence, the relative energies of the TSs increase by between 2.5 and 5.9 kcal·mol^−1^. In methanol, the activation energy increases by 0.9 kcal·mol^−1^, and the 32CA reaction remains completely regio- and stereoselective, as **TS-mn** and **TS-ox** are found to be 4.0 and 5.0 kcal·mol^−1^, respectively, above **TS-on**. The 32CA reaction remains strongly exothermic by −45.2 kcal·mol^−1^.

The *ω*B97X-D/6-311G(d,p) thermodynamic data of the 32CA reaction of AY **21** with ferrocene ethylene **22** were further analyzed. The relative enthalpies, entropies, and Gibbs free energies are given in Table 3, while the thermodynamic data are given in Appendix A in Appendix A.

A representation of the enthalpy and Gibbs free energy profiles associated with the four competitive reaction paths is given in Figure 3. The inclusion of the thermal corrections to the electronic energies in methanol increases the relative enthalpies only by between 0.8 and 3.4 kcal·mol^−1^. Indeed, they have a markedly low incidence in the relative enthalpies of the TSs, which only increase by between 0.8 and 1.1 kcal·mol^−1^ with respect to the electronic energies in methanol. Considering the activation enthalpies, **TS-ox** and **TS-mx** are found to be more than 3.7 kcal·mol^−1^ higher in enthalpy than **TS-on**. The inclusion of entropies to enthalpies increases the relative Gibbs free energies by between 16.1 and 20.3 kcal·mol^−1^ as a consequence of the unfavorable activation entropies associated with these bimolecular processes, which are found in the range −47.6 and −60.2 cal·mol^−1^·K^−1^, and the temperature of the reaction of 65 °C. The formation of **MC-on** is exergonic by −2.5 kcal·mol^−1^. The activation Gibbs free energy associated with this 32CA reaction via **TS-on** rises to 12.6 kcal·mol^−1^, while the formation of spirooxindole **23** is strongly exergonic by −22.4 kcal·mol^−1^.

It is interesting to highlight that while the activation enthalpies suggest that this 32CA reaction is completely *ortho* regioselective, in full agreement with the experimental outcomes [45], the activation Gibbs free energies markedly decrease the *ortho* regioselectivity as **TS-mn** is only 0.5 kcal·mol^−1^ above **TS-on**. This behavior is a consequence of the more unfavorable activation entropy associated with the *ortho* TSs (see later).

The geometries of **MC-on** and the four TSs are given in Figure 4 and Figure 5, respectively. At **MC-on**, the two interacting frameworks, which are separated by a distance of ca. 3.11 Å, present a parallel rearrangement (see Figure 4). At the more favorable *ortho* TSs, the distances between the two pairs of C3–C4 and C1–C5 interacting carbons are 2.112 and 2.704 Å, respectively, at **TS-on** and 2.126 and 2.562 Å, respectively, at **TS-ox**, while at the *meta* TSs, the distances between the two pairs of C1–C4 and C3–C5 interacting carbons are: 2.081 and 2.633 Å, respectively, at **TS-mn** and 2.320 and 2.268 Å, respectively, at **TS-mx** (see Figure 5). These distances indicate that while the most favorable **TS-on** shows a high geometrical asynchronicity with ∆l = 0.59 Å, the most unfavorable **TS-mx** shows a very low geometrical asynchronicity with ∆l = 0.05 Å. The most favorable highly asynchronous **TS-on** is associated with a two-center interaction between the C3 carbon of AY **21**, the second most nucleophilic center of this TAC, and the *β*-conjugated C4 carbon of ferrocene ethylene **22**, the most electrophilic center of this ethylene derivative.

A detailed analysis of the geometry of the most favorable **TS-on** shows that one of the hydrogens of the ferrocene framework of ethylene **22** is located at 2.439 Å of the carbonyl oxygen atom of AY **21**, and one hydrogen of the dihydropyrrole ring of AY **21** is located at 2.418 Å of the carbonyl oxygen atom of ferrocene ethylene **22** (see Figure 5). These distances suggest the presence of two hydrogen bonds (HBs) between the hydrogen and oxygen centers. Interestingly, these HBs which are already present at the most stable **MC-on**, with H–O distances of 2.341 and 2.377 Å, respectively (see Figure 4), may account for the *ortho* and *endo* selectivity found in this 32CA reaction. The formation of these HBs, with the first one exhibited by a slight twist of the cyclopentadienyl ring with respect to the C4–C5 double bond, could be responsible for the lower entropy associated with **TS-on**, i.e., 230.2 cal·mol^−1^·K^−1^, than that of **TS-mn**, i.e., 239.7 cal·mol^−1^·K^−1^ (see Appendix A in Appendix A) and, consequently, for the loss of *ortho* regioselectivty when relative Gibbs free energies are considered (see Table 3).

On the other hand, the analysis of the geometry of the regioisomeric **TS-mn** suggests the presence of only one HB between the carbonyl O7 oxygen of ferrocene ethylene **22** and one of the dihydropyrrole hydrogen of AY **21**, with a H–O distance of 2.337 Å (see Figure 5). Thus, the presence of an additional HB at the most favorable **TS-on** accounts for the change of regioselectivity in this 32CA reaction.

Finally, the analysis of the GEDT [26] at **TS-on** permits the assessment of the polar character of this 32CA reaction. GEDT values lower than 0.05 e correspond to non-polar processes, while GEDT values higher than 0.20 e correspond to high polar processes. The GEDT value at the two stereoisomeric *ortho* TSs is 0.23 e. This high value is a consequence of the supernucleophilic character of AY **21** and the strong electrophilic character of ferrocene ethylene **22**. The flux of the electron density, which goes from AY **21** to ferrocene ethylene **22**, allows classifying this 32CA reaction as FEDF, in clear agreement with the analysis of the CDFT indices. The high polar character of this 32CA reaction accounts for the fact that, considering the relative enthalpies, **TS-on** is located below the separated reagents (see Table 3) [60].

Figure 6 shows the ELF [46] basin attractor positions of **MC-on** and **TS-on**. The ELF of **MC-on** presents similar features to those of the separate reagents, i.e., AY **21** and ferrocene ethylene **22** (see Figure 1). The AY framework of **MC-on** shows the presence of one V(C1) monosynaptic basin, integrating 0.33 e, already present at *pseudo(mono)radical* AY **1**, while the C4–C5 double bond of the ferrocene ethylene moiety is characterized by the presence of two disynaptic basins, V(C4,C5) and V’(C4,C5), integrating a total of 3.27 e (see Figure 1 and Figure 6).

The most relevant feature of **TS-on** is the creation of a new V(C3) monosynaptic basin integrating 0.49 e, while the V(C1) monosynaptic basins present at **MC-on** is slightly depopulated to 0.29 e. This new monosynaptic basin, which is created at the second most nucleophilic center of AY **21** (see Parr functions in Figure 2), is demanded for the subsequent creation of the first new C3–C4 single bond [26]. On the other hand, the two disynaptic basins associated with the C4–C5 partial double bond present at ferrocene ethylene **22** and **MC-on** merge into one single V(C4–C5) disynaptic basin at **TS-on** after losing 0.71 e from ethylene **22**. This ELF analysis of **TS-on**, which accounts for the non-concerted nature of this one-step 32CA reaction, supports the previous analysis based on the geometrical parameters.

### 2.4. Origin of the Ortho/Endo Regioselectivity

As has been aforementioned, the geometries of the more favorable **MC-on** and **TS-on** suggest the presence of two HBs between the carbonyl oxygen atoms and the ring hydrogen atoms, which might be responsible for the unexpected *ortho* regioselectivity (see Figure 4 and Figure 5). Thus, in order to confirm their presence, a topological analysis of the electron density associated with the intermolecular non-covalent interactions (NCI) taking place at both stationary points was performed by means of the Independent Gradient Model [61,62] (IGM). The corresponding isosurfaces are represented in Figure 7.

IGM-δg_inter_ at **MC-on** shows a green surface between the main interacting regions of both AY and ethylene frameworks and two additional green surfaces involving the two carbonyl oxygen atoms and one of the cyclopentadienyl or dihydropyrrole hydrogen atoms. While the more extended surface is related to the non-covalent interactions in the bond formation region, which turns to a blue-to-red color at **TS-on** as a consequence of the higher proximity between the two frameworks and the more advanced C3–C4 bond formation, the two surfaces in the O–H regions are related to weak HBs with δg_inter_ signatures and instrinsic bond strength index (IBSI) [63] values of ca. 0.03 a.u. These surfaces become smaller, and the respective HBs weaker, at **TS-on** due to a slight elongation of the corresponding O–H distances by 0.065 and 0.013 Å. Although their contributions to the intermolecular interactions [64] are lower than 4%, with the O(Et)–H(AY) interaction contributing 0.6% more than the O(AY)–H(Et) interaction, their presence seems to outweigh the more favorable electrophilic/nucleophilic electronic interactions involving the most nucleophilic C1 *pseudoradical* center along the *meta*/*endo* path [24], thus leading to the *ortho*/*endo* selectivity.

Finally, the IGM-δg_inter_ at the regioisomeric **TS-mn** shows the presence of only one HB between the carbonyl O7 oxygen of ferrocene ethylene **22** and one of the dihydropyrrole hydrogens of AY **21** (see Appendix A in Appendix A). Consequently, the additional HB present at the most favorable **TS-on** can explain the fact that this TS is 5.4 kcal·mol^−1^ more stable than **TS-mn**, and, consequently, the origin of the unexpected *ortho* regioselectivity of this 32CA reaction [45].

## 3. Materials and Methods

The *ω*B97X-D [65] functional, together with the standard 6-311G(d,p) basis set [66], which includes d-type polarization for second-row elements and p-type polarization functions for hydrogen atoms, were used in this MEDT study. The TSs were characterized by the presence of only one imaginary frequency. The Berny method was used in optimizations [67,68]. The IRC [69] calculations were performed to establish the unique connection between the TSs and the corresponding minima [70,71] phase structures at the same computational level using the polarizable continuum model (PCM) [72,73] in the framework of the self-consistent reaction field (SCRF) [74,75,76]. Values of *ω*B97X-D/6-311G(d,p) enthalpies, entropies, and Gibbs free energies in methanol were calculated with standard statistical thermodynamics at 337.8 K and 1 atm [66], by PCM frequency calculations at the solvent optimized structures.

The GEDT [26] values were computed by using the equation GEDT(f) = Σq_f_, where q are the natural charges [48,49] of the atoms belonging to one of the two frameworks (f) at the TS geometries. Global and local CDFT indices [50,51] were calculated by using the equations given in reference [51], using the B3LYP/6-31G(d) method, because the original nucleophilicity and electrophilicity scales were established at that level [51].

The Gaussian 16 suite of programs was used to perform the calculations [77]. ELF [46] analyses of the *ω*B97X-D/6-311G(d,p) monodeterminantal wavefunctions were performed by using the TopMod [78] package with a cubical grid of step size of 0.1 Bohr. Molecular geometries and ELF basin attractors were visualized by using the GaussView program [79]. IGM analysis was carried out with the IGMPlot software [62].

## 4. Conclusions

The 32CA reaction of AY **21**, derived from isatin and L-proline, with ferrocene ethylene **22**, yielding spirooxindole **23**, has been studied within MEDT at the *ω*B97X-D/6-311G(d,p) computational level. Analysis of the ELF topology of AY **21** indicated that this TAC has a *pseudo(mono)radical* structure characterized by the presence of two monosynaptic basins, integrating a total of 0.77 e, at the C1 carbon. The analysis of the CDFT reactivity indices indicated that ferrocene ethylene **22** has a strong electrophilic characteristic, while AY **21** is a supernucleophile, suggesting that the corresponding 32CA reaction has a high polar character. The most favorable reaction path via the *ortho*/*endo* **TS-on** presents an activation enthalpy of 8.7 kcal·mol^−1^, with the 32CA reaction being strongly exothermic by −42.1 kcal·mol^−1^. Analysis of the activation enthalpies indicated that this reaction presents a complete *endo* stereoselectivity and a complete *ortho* regioselectivity, in agreement with the experimental outcomes. Analysis of the GEDT at the most favorable **TS-on**, 0.23 e, accounts for the high polar character of this 32CA reaction, classified as FEDF. The presence of two HBs between the two carbonyl oxygens and the two ring hydrogens at the most favorable **TS-on**, which are already present at **MC-on**, appear to be responsible for the unexpected *ortho* regio- and *endo* stereoselectivity found in this 32CA reaction involving the *pseudo(mono)radical* AY **21**.

## Data Availability

Not applicable.

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
