# Peer review of "A Molecular Electron Density Theory Study of the [3+2] Cycloaddition Reaction of an Azomethine Ylide with an Electrophilic Ethylene Linked to Triazole and Ferrocene Units"

_molecules, 2022, doi:10.3390/molecules27196532_

Round 1

Reviewer 1 Report

This paper reports an in depth theoretical study of a 3+2 cycloaddition with the main goal of explaining his unusual ortho regioselectivity. The mechanism for the cycloaddition was identified as pmr-type with high polar character and the possibile reaction pathways were analyzed in order to find the most energetically favored one. The formation of H-bonded intermolecular interactions between the two reagents in the initial adduct MC-on were proposed to favor the experimentally observed selectivity.

In my opinion this article is a well structured and interesting study, I would recommend his publication in Molecules after some adjustments. In particular, it seems to me there are some inconsistencies between the results and the conclusions, which I explain as follows:

1) The Gibbs free energy activation barriers of the four possible pathways are not enough different (0.5 kcal/mol between TSon and TSmn) to be in agreement with the experimentally observed complete regio and stereoselectivity. However, the authors states in the conclusions that "Analysis if the activation enthalpies indicates that this reaction presents a complete endo stereoselectivity and a complete ortho regioselectivity, in agreement with the experimental outcomes". This is controversial. In my opinion considering only the activation enthalpy instead of the Gibbs free energy of activation to show an agreement with the experimental data would deserve a further explanation.

2) While ELF and CDFT analysis point towards the formation of a meta/endo product, the authors suggest that the presence of hydrogen bonding (HB) in MC-on outweighs the electrophilic-nucleophilic interactions to give the observed ortho/endo product. These was based on the O--H distances in the optimized MC-on structure and IGM. Since these HBs involves scarsely polarized C-H bonds, it is sounds strange for these interactions to be the driving force of the process selectivity. Especially in methanol, which is the actual reaction solvent. I think some more "quantitive" evidences would be necessary to prove the author hypothesis. For instance, How much MC-on is the most favoured 21-22 adduct in energy terms?

Other minor observations are listed below:

- in line 29 I think the word "enantiomerically" is mispelled

- in line 56 and 61 there is a reference to compounds 7 and 8 whose structures do not appear in the main text. Maybe an additional figure is required.

- in Scheme 4 I guess the absolute configuration of the stereocenter at C3 is missing

- in Figure 1 I find the color scheme for the atom charges a bit confusing, since it changes apparently regardless of charge sign and intensity. Would it be more clear if negative charges were put in one colour and all positive charges with another one?

- in Figure 3 I think the product label are necessary also for the enthalpy diagram

- in line 344 "non-concerted nature of this one-step 32CA reaction". How can a one-step cycloaddition be non-concerted? Can the authord clarify this point?

Author Response

In my opinion this article is a well structured and interesting study, I would recommend his publication in Molecules after some adjustments. In particular, it seems to me there are some inconsistencies between the results and the conclusions, which I explain as follows:

1) The Gibbs free energy activation barriers of the four possible pathways are not enough different (0.5 kcal/mol between TSon and TSmn) to be in agreement with the experimentally observed complete regio and stereoselectivity. However, the authors states in the conclusions that "Analysis if the activation enthalpies indicates that this reaction presents a complete endo stereoselectivity and a complete ortho regioselectivity, in agreement with the experimental outcomes". This is controversial. In my opinion considering only the activation enthalpy instead of the Gibbs free energy of activation to show an agreement with the experimental data would deserve a further explanation.

R. We are in complete agreement with the referee’s comment. With our large experience in the field of the study of chemical organic reactivity, in general the analysis of the relative enthalpies gives better results than the analysis of the Gibbs free energies, and this is a problem of the computation of the relative entropies.

Our computational models can never reproduce the experimental outcomes, where there is a very high number of interacting molecules in solution. Although the feasible interactions of the reagents with other molecules, including solvent molecules, can modify the relative enthalpies, they have a more decisive role in the calculation of relative entropies, and consequently, it is manifested in the relatives Gibbs free energies, which sometimes do not reproduce the experimental outcomes. Part of this behavior has been emphasized in the text of the revised version.

2) While ELF and CDFT analysis point towards the formation of a meta/endo product, the authors suggest that the presence of hydrogen bonding (HB) in MC-on outweighs the electrophilic-nucleophilic interactions to give the observed ortho/endo product. These was based on the O--H distances in the optimized MC-on structure and IGM. Since these HBs involves scarsely polarized C-H bonds, it is sounds strange for these interactions to be the driving force of the process selectivity. Especially in methanol, which is the actual reaction solvent. I think some more "quantitive" evidences would be necessary to prove the author hypothesis. For instance, How much MC-on is the most favoured 21-22 adduct in energy terms?.

R. After many studies of HB stabilized TSs, these favorable interactions are not only geometrically characterized by the H-O distance, which should be below 2.5 Å, but also by the rearrangement of the corresponding hydrogen with respect to the oxygen atom. Once these geometric features are recognized, the HBs are topologically characterized by IGM analysis.

We have optimized the molecular complex MC-mn, and in gas phase it is 5.1 kcal/mol less stabilized than MC-on, an energy difference similar to that between TS-on and TS-mn. In addition, in this revised version, we have performed an IGM analysis of TS-mn, showing the presence of only one HB (see the new Figure S1 in Supplementary Material). The energy difference between the two regioisomeric endo TSs, 5.5 kcal/mol, which is similar to that between the MCs, is in agreement with the presence of one additional HB at the most favorable ortho TS-on.

Other minor observations are listed below:

- in line 29 I think the word "enantiomerically" is misspelled

R. The corresponding misspell has been corrected.

 - in line 56 and 61 there is a reference to compounds 7 and 8 whose structures do not appear in the main text. Maybe an additional figure is required.

R. The molecular formula of the simplest azomethyne ylide AY 7 CH2-NH-CH2 has been written in line 55 in this revised version. That of the simplest carbonyl ylide 8 CH2-O-CH2 was already shown in line 56. These simplest TACs do not need an additional figure.

- in Scheme 4 I guess the absolute configuration of the stereocenter at C3 is missing.

R. In agreement with the referee’s suggestion, the absolute configuration of the stereocenter at C3 has been indicated in spirooxindole 20 and 23 in Scheme 4.

- in Figure 1 I find the color scheme for the atom charges a bit confusing, since it changes apparently regardless of charge sign and intensity. Would it be more clear if negative charges were put in one colour and all positive charges with another one?.

R. These colors have always been used in our manuscripts to represent the sign of the charges. They have never been questioned. Red is for negative charges, blue for positive, and green for neutral (which are lower than 0.1 in absolute value). In any case, we have put the values in bold to make them more clearly visible.

in Figure 3 I think the product label are necessary also for the enthalpy diagram

R. In agreement with the referee’s suggestion, the labels of all species have been included in enthalpy diagram in Figure 3.

- in line 344 "non-concerted nature of this one-step 32CA reaction". How can a one-step cycloaddition be non-concerted? Can the author clarify this point?

R. Many MEDT studies have proved that the concerted mechanism for cycloaddition reactions does not exit. They are simply reactions taking place via a one-step mechanism. See the Eur. J. Org. Chem. 2018, 1107 entitled “The Mysticism of Pericyclic Reactions. A Contemporary Rationalisation of Organic Reactivity Based on the Electron Density Analysis”.

Reviewer 2 Report

The authors performed a theoretical study of the reaction mechanism between Azomethine Ylide and an ethylene with a ferrocene as functional group. This particular [3+2] cycloadition produce the unexpected ortho product instead of the meta. They use several electron density indices to classify the type of transition states involved in the reaction mechanism and to explain the opposite selectivity. The manuscript is well written and the methodology is adequate. I only have one concern.

To explain the regioselectivity, they use IGM isosurfaces to find important non-covalent interactions, but they only analyze the endo, ortho reactive dimer (MC-on) and TS-on. In order to really probe that the non-covalent interactions are responsible for the observed selectivity, an IGM isosurface of the other dimer should be calculated, at least the (MC-mn) dimer that is the most probable that forms before the TS-mn. This TS is the second with the lowest activation energy and would produce the expected regioisomer. If no other prereaction dimer is formed, the IGN analysis should be done in the TSs.

Minor issues

Scheme 3 in compound 15 on the spiro carbon there are 2 hatched bonds and should be one.

In page 5 line 147. I think that it should be 1.0 e instead of 0.1e

In page 5 line 161. Should be C6 instead of C5.

In Figure 3 the names of the TS and the products are not indicated in the enthalpy paths.

Author Response

To explain the regioselectivity, they use IGM isosurfaces to find important non-covalent interactions, but they only analyze the endo, ortho reactive dimer (MC-on) and TS-on. In order to really probe that the non-covalent interactions are responsible for the observed selectivity, an IGM isosurface of the other dimer should be calculated, at least the (MC-mn) dimer that is the most probable that forms before the TS-mn. This TS is the second with the lowest activation energy and would produce the expected regioisomer. If no other prereaction dimer is formed the IGN analysis should be done in the TSs.

R. In agreement with the referee’s suggestion, the IGM analysis of TS-mn has been carried out in this revised version. It shows the presence of only one HB at this TSs (see the new Figure S1 in Supplementary Material). Consequently, the presence of one additional HB at the most favorable ortho TS-on accounts for the additional stabilization by 5.5 kcal/mol of this TS with respect to TS-mn.

Minor issues

Scheme 3 in compound 15 on the spiro carbon there are 2 hatched bonds and should be one.

R. In agreement with the referee’s suggestion, the corresponding mistake has been corrected.

In page 5 line 147. I think that it should be 1.0 e instead of 0.1e

R. Thanks, the mistake has been corrected.

 In page 5 line 161. Should be C6 instead of C5.

R. The corresponding mistake has been corrected.

In Figure 3 the names of the TS and the products are not indicated in the enthalpy paths.

R. In agreement with the referee’s suggestion, the labels of all species have been included in enthalpy diagram in Figure 3.

Reviewer 3 Report

The molecular electron density theory (MEDT), which has been raised by the authors, was used for understanding the 1,3-dipolar cycloaddition of an azomethine ylide with the ethylene linked to ferrocene and triazole fragments in this manuscript. To be specific, wavefunction analysis, CDFT studies, and energy evaluations were used for understanding this cycloaddition. These researches are quite excellent. Before being accepted by Molecules, I would like to ask some small questions/suggestions about this manuscript:

1.  I am interested in the electronic structure of azomethine ylides (AYs) as a 1,3-dipole compound. The authors use "pseudo(mono)radical TAC" to describe its electronic structure, I am curious why the radical feature is used to describe a molecule with a closed shell. Does this molecule have a low excitation energy or degenerated electronic states? Perhaps I can understand the authors' intention, indeed, there may be the potential mixing from several excited states when a cycloaddition occurs, however, azomethine ylides should be diamagnetic compounds. Additionally, the "diradical" configuration of AYs might be obtained in my opinion, rather than "radical" (two unpaired electrons locate at two carbon atoms?).

2. The chemical structure in Scheme 1 and Scheme 2 should be corrected. Of course, it is possible to use resonance formulas (with formal charges) or delocalized representations.

3. Adding a comparison of secondary interaction (NCI) analysis among several different transition states (TS-ox, TS-mx, TS-mn, TS-on) in Figure 7 would be better.

4. The authors are recommended to add the original calculated data to the Supplementary Information, such as the coordinates of intermediates and transition states in the manuscript, etc.

Author Response

Before being accepted by Molecules, I would like to ask some small questions/suggestions about this manuscript:

- I am interested in the electronic structure of azomethine ylides (AYs) as a 1,3-dipole compound. The authors use "pseudo(mono)radical TAC" to describe its electronic structure, I am curious why the radical feature is used to describe a molecule with a closed shell. Does this molecule have a low excitation energy or degenerated electronic states? Perhaps I can understand the authors' intention, indeed, there may be the potential mixing from several excited states when a cycloaddition occurs, however, azomethine ylides should be diamagnetic compounds. Additionally, the "diradical" configuration of AYs might be obtained in my opinion, rather than "radical" (two unpaired electrons locate at two carbon atoms?).

R. The concept of pseudoradical species was introduced in 2012 by Domingo to describe closed shell molecules in which the topological analysis of the electron density shows the presence of a non-bonding region at a carbon atom integrating less than 1.0 e (see J. Org. Chem. 2011, 76, 373). Note that carbenoid species, which are also closed shell molecules, have a non-bonding region in a carbon atom integrating near 2.0 e.

ELF analysis of the simplest AY CH2-NH-CH2 shows the presence of a pseudoradical center at each of the two terminal carbons (see Understanding the High Reactivity of the Azomethine Ylides in [3 + 2] Cycloaddition Reactions. Lett. Org. Chem.2010,7,432). Thus, in 2014, the simplest AY CH2-NH-CH2 was classified as a pseudodiradical TAC (see Understanding the mechanisms of [3+2] cycloaddition reactions. The pseudoradical versus the zwitterionic mechanism in Tetrahedron,2014,70,1267).

The ELF analysis of the AY used in this work presents only one pseudoradical center due to the substitution, therefore it is classified as a pseudo(mono)radical TAC, instead of a pseudodiradical.

-The chemical structure in Scheme 1 and Scheme 2 should be corrected. Of course, it is possible to use resonance formulas (with formal charges) or delocalized representations.

R. Our extensive study in the field of 32CA reactions have shown that most of the TACs cannot be represented neither by a single Lewis structure nor by a resonant structure. In addition, the formal charges do not represent the actual charge distribution (see the charge distribution of AY 21 in Figure 1 in which the two carbons are negligibly charged).

In addition, the formal charges are very confusing for organic chemists as they can suggest an erroneous nucleophilic/electrophilic local activation (see Unravelling the Mysteries of the [3+2] Cycloaddition Reactions. Eur. J. Org. Chem. 2019, 267)

After the analysis of the electronic structure of AY 21, its pseudomonoradical nature has been represented in Figure 1 and Scheme 4.

- Adding a comparison of secondary interaction (NCI) analysis among several different transition states (TS-ox, TS-mx, TS-mn, TS-on) in Figure 7 would be better.

R. In agreement with the referee’s suggestion, in this revised version we have studied the IGM of the meta/endo TS-mn. This analysis shows the presence of only one HB at TS-mn (see the new Figure S1 in Supplementary Material). Consequently, the presence of one additional HB at the most favorable ortho TS-on accounts for the additional stabilization by 5.5 kcal/mol of this TS with respect to TS-mn.

- The authors are recommended to add the original calculated data to the Supplementary Information, such as the coordinates of intermediates and transition states in the manuscript, etc.

R. In agreement with the referee’s suggestion, the Cartesian coordinates of all stationary points have been included in the Supplementary Information in this revised version.

Round 2

Reviewer 1 Report

I think the authors have addressed my two main observations properly.  Concerning point 2, I agree that the lack of one HB interaction in TS-mn can be an indication of the complete ortho/endo regioselectivity.

Corrections are fine for the other minor issues.